# Using Panel Data to Evaluate the Factors Affecting Transport Energy Consumption in China’s Three Regions

**DOI:** 10.3390/ijerph16040555

**Published:** 2019-02-14

**Authors:** Tianxiang Lv, Xu Wu

**Affiliations:** School of Traffic and Transportation, Beijing Jiaotong University, Beijing 100044, China; 17120854@bjtu.edu.cn

**Keywords:** transportation industry, energy consumption, influencing factors, STIRPAT model, panel data

## Abstract

In China, transportation accounts for a large proportion of total energy consumption and that trend is projected to increase in the future. Through the stochastic impacts by regression on population, affluence, and technology (STIRPAT) model, OLS regressions were conducted to investigate the impacts of gross domestic product (GDP), urbanization, energy intensity and transport structure on the transport energy consumption in China’s three regions. The analyses of inter-provincial panel data during the period 2006–2015 is compared to the analysis of the data from 1996 to 2005 to determine the change. There were two primary findings from this study. First, the changes of the influencing degree in three regions are considered. GDP is still the main driver of transport energy consumption in eastern region, while urbanization becomes the main driver in the other two regions. Second, the relationship between the elasticity and the value of each variable is detected. The elasticity of transport energy consumption with respect to GDP, transport structure, energy intensity and urbanization have separate positive and significant relationships. The primary measure is to optimize transport structure in the central region, while reducing energy intensity in the western region. Finally, we propose relevant policy recommendations for the three regions.

## 1. Introduction

China’s transportation industry has flourished since 2006 as a result of industrialization and urbanization. Data from the National Bureau of Statistics (NBS) indicates that from 2006 to 2015, China’s passenger-km increased by 56.6% and the freight ton-km doubled. 

Along with active transport activities, transport energy consumption has also been rapidly increasing. Since 2010, China has surpassed the United States to become the world’s largest energy consumer [1]. According to the International Energy Agency (IEA), by 2035, the energy consumption of the world’s transportation sector will increase by 43% and China will account for more than one-third of the energy consumption [2]. The rapid increase in the transport energy consumption will result in serious energy security problems and environmental pressures in China. The energy outlook 2018 released by BP shows that oil import dependence will increase from 63% in 2016 to 72% in 2040. Gas dependence will increase from 34% to 43% in 2040 [3]. Therefore, it is vital to evaluate the factors affecting transport energy consumption to clarify futher steps for policy makers.

The NBS provides a division of three regions based on the economic development level and geographic location of each province, as shown in Figure 1. In 2006, 197 million tons of standard coal were used; by 2015, the use had increased by 63.2% in the entire country. The features of energy consumption in the transport sector are also different among regions, as shown in Figure 2. Among them, Qinghai, Ningxia, and Hainan in the western region exhibit fewer changes and remain at the lowest energy consumption level. The increase in energy consumption is more apparent in the eastern and central regions. The four provinces in the highest energy consumption level in 2015 are all in the eastern region. 

Moreover, development policies, technological levels, and industrial structures also differ among the three regions in China and previous studies have seldom considered the differences between regions. The results obtained from the analysis of the entire country is not always applicable to a specific region. Most previous studies have focused on a particular transport mode and few studies have focused on the analysis of the entire transportation sector. Decomposition and the grey system theory (GST) method have been commonly used, but econometric methods have not been widely applied. 

This paper aims to: (a) obtain the factors affecting transport energy consumption using the extended stochastic impacts by regression on population, affluence, and technology (STIRPAT) model; (b) investigate the level of influence of various factors in the three regions between 2006 and 2015; and (c) evaluate the relationship between the elasticity and the value of each variable by comparing the results to those from 1996 to 2005. The results show that the gross domestic product (GDP) is the main driver of transport energy consumption in the eastern region, while urbanization is the main driver in the other two regions. At present the primary measure to reduce transportation energy consumption is optimizing transport structure, rather than reducing energy intensity in the central regions. Based on the results, we propose relevant policy recommendations in three regions.

## 2. Methodology and Data

The ordinary least squares (OLS) method finds the best function match of the data by minimizing the sum of the squared errors. The least squares estimate is the best linear unbiased estimate of the model parameters. The OLS approach is applied based on time series data, cross section data or panel data. Most studies have analyzed time series data and cross section data, whereas the use of panel data is less common. Alcaraz et al. [4] used time series data to study the impact of transport infrastructure on environmental change. Zhang et al. [5] conducted a regression analysis of provincial panel data in China from 1995 to 2010. They identified the degree of influence of various factors. The results showed that energy efficiency was effective, but it had a limited impact on reducing emissions. Unlike the other two types of data, panel data take into account heterogeneity and avoid the multicollinearity that often exists in time series data [6]. Moreover, the heterogeneity and time-varying features can be considered together to obtain the most comprehensive features of the region and, thus, the most accurate results.

The independent variables need to be determined first for OLS regression analysis. There are four main methods applied to obtain the factors affecting transportation energy consumption: Kaya equation; ASIF equation; impacts by regression on population, affluence, and technology (IPAT) model; and the STIRPAT model. The Kaya equation [7] converts the interpreted variables into the product of multiple influencing factors. It requires that there must be a mutual transformation relationship between the explanatory variables. The ASIF method [8] only considers the factors of activity level, traffic structure, energy intensity, and energy use. The IPAT model [9] only considers three factors and does not allow hypothesis testing [10]. The selection of factors is more comprehensive in the STIRPAT model [11] because the population, affluence, and technology can be decomposed into a number of factors affecting the environment. In addition, it can better reflect the non-monotonic or non-proportional function relationship among the influencing factors [12]. Therefore, the STIRPAT model is used to determine the factors. The influencing factors considered are *GDP*, urbanization (*U*), energy intensity (*EI*) and transport structure (*TS*) as discussed in Section 3.2.

Many scholars have discussed the impact of the variables on transport energy consumption, respectively on energy consumption in general or on emissions. Achour et al. [13] estimated the impact of transport structure on transport energy consumption in Tunisia. The authors concluded that the effect of transport structure on energy consumption is positive in Tunisia where road transport assumes an important proportion of transportation activity. Zhang et al. [14] found that the ratio of energy consumption to transport ton-km (*TO*) of each transport mode plays the dominant role in decreasing transport energy consumption in China. Liu et al. [15] noted that *GDP* was the primary driving factor affecting transport energy consumption. In addition, *EI* (ratio of energy consumption to transport ton-km) was the primary factor affecting the reduction of CO_2_ emissions from road transport. Yuan et al. [16] determined the different trends of *EI* (ratio of energy consumption to transport ton-km) of different provinces. Lin et al. [17] determined that a reduction in the *EI* was beneficial to China’s transportation industry. *GDP* growth is the second largest influencing factor of CO_2_ emissions in the transport industry. *U* has a positive effect on transport energy consumption.

Xu et al. [18] found that the nonlinear impact of urbanization exhibits an inverted “U-shaped” pattern in China’s transport sector. The nonlinear effect of economic growth on CO_2_ emissions is consistent with the Environmental Kuznets Curve (EKC) hypothesis, whereas inconsistent findings have also been detected [19]. Therefore, further validation of the EKC hypothesis in China is imperative. 

The existing literatures analyzed the impacts of variables on transport energy consumption from the perspective of the whole country, while the differences of the impacts are not examined between regions. The analysis of the change of the degree of influence is rare. Therefore, we conduct a regional analysis of the factors influencing transport energy consumption in China. We obtain the most accurate estimates using panel data. The research results will provide new insights into the energy conservation and emission reduction of the transportation industry in different regions of China.

### 2.1. STIRPAT Model

The STIRPAT model was first proposed by Dietz and Rosa [7] and is often used to analyze factors influencing the environment. The model is an improvement of the IPAT model proposed by Ehrlich and Holdren [20]. The model uses different weights that represent the degree of influence of the factors in the form of indices; thereby overcoming the disadvantage of the same degree of influence of various factors in the IPAT model. Therefore, the STIRPAT model well reflects the non-proportional functional relationships among various factors [20]. The model is shown in Equation (1):(1)$${I\;=\;\mathrm{aP}}^{b}A^{c}T^{d}e$$
where *I* denotes the effects of human activities on the environment, *P* represents the population, *A* stands for affluence, and *T* represents technology level. *a* is the coefficient of the model; *e* is the error term; and *b*, *c*, and *d* are the exponents of *P*, *A*, and *T*, respectively. When *a* = *b* = *c* = *d* = *e* = 1, the equation describes the IPAT model.

### 2.2. Extended STIRPAT Model

Urbanization in this paper is measured by the proportion of the urban population in total population. China is currently in the process of urbanization. It is predicted that 60% of the population will be living in cities by 2020. It is well known that the secondary and tertiary industries are the main factors affecting the flow of passengers and cargo and the population engaged in the secondary and tertiary industries is mainly concentrated in cities. The characteristics of urban areas determine the travel distance and the transportation modes [21], which ultimately affect energy consumption. The population in urban area causes the main part of transport energy consumption. Using total population is not suitable, therefore *U* is used as an influencing factor to reflect the population which influences the environment actually.

In current studies, the *GDP* or per capita *GDP* are commonly used to represent the level of wealth in a region. Wu et al. [22] used the *GDP* as a factor affecting the freight transport energy consumption. Shuai et al. [23] selected the per capita *GDP* as an influencing factor of transport energy consumption in Xinjiang. The *GDP* reflects the overall level of wealth in a region, whereas the per capita *GDP* represents the average level of wealth. Moreover, the *GDP* directly determines the transportation demand [24]; therefore, the effects of the *GDP* on transport energy consumption are investigated.

In the STIRPAT model, *P*, *A*, and *T* can be decomposed into multiple influencing factors affecting the environment [25]. Based on the results of previous studies, we use the *EI* and *TS*, which is the shares of road transport to reflect the technology level.

There are two main methods to calculate *EI*. One method uses the ratio of transport energy consumption to the *GDP* in the transportation sector and the other method uses the ratio of the transport energy consumption to the transport ton-km. The transport ton-km is considered the most suitable indicator for measuring the total output of the transportation industry [26]; therefore, the second method accurately reflects the energy efficiency of the transport activity and it is used in this study. Wang et al. [27] used this method and the LMDI method to study the influence of the *EI* on the transport CO_2_ emissions. The empirical results showed that road transport was the main contributor to carbon emissions related to energy consumption in China. Therefore, the transport structure in this study is expressed by the proportion of the road transport ton-km to the total transport ton-km.

Regardless of the measuring methods, the estimation results remain the same in the logarithmic model that measures the change in percentage. At the same time, the logarithmic form eliminates the possible heteroscedasticity in the data so that better estimation results are obtained [28]. Since e and a represent the random errors and constant, respectively, their logarithmic form cannot be derived. Therefore, the extended STIRPAT model is used, as shown in Equation (2):(2)lnE = a + blnU + clnGDP + dlnEI + flnTS + e
where *E* represents the transport energy consumption, *U* represents the urbanization rate, *GDP* is the regional *GDP*, *EI* represents the energy intensity, and *TS* represents the transport structure. *b*, *c*, *d*, and *f* indicate that the transport energy consumption will increase by *b*%, *c*%, *d*%, and *f*% when there is a 1% increase in the corresponding variables. The representations and meanings of the above variables are shown in Table 1.

To calculate *EI* and *TS*, we should first obtain the value of transport ton-km, as shown in Table 1. The transport ton-km is calculated using the method proposed by Zhou et al. [29] (transport ton-km volume = passenger ton-km volume × passenger ton-km conversion coefficient of each mode + freight ton-km volume). The coefficients are determined by comparing the energy consumption of one ton of cargo per km to the energy consumed to transport one passenger per km. The coefficients of the inland waterways, domestic aviation, highways, and railways are 0.3, 0.072, 0.1, and 1 respectively. The value of transport ton-km is adjusted, as shown in Appendix A. Therefore *EI* and *TS* can be obtained as shown in Equation (3) and Equation (4), respectively:(3)$$\mathrm{EI}\;=\;\frac{E}{\mathrm{TO}}$$
(4)$$\mathrm{TS}\;=\;\frac{{\mathrm{TO}}_{\mathrm{road}}}{\mathrm{TO}}\;$$
where *EI* represents energy intensity, *E* represents the transport energy consumption, *TO* represents transport ton-km, *TS* represents transport structure, and *TO_road_* represents road transport ton-km. 

### 2.3. Establishment of Panel Data Regression Model

Panel data regression models were first used by Balestra and Nerlove [31] for economic analysis. This method reduces the influence of inter-variable collinearity, controls individual heterogeneity, and increases the degrees of freedom, thereby improving the estimation efficiency [32]. Panel data regressions differ from time series regressions or cross-sectional regressions in that they have double subscripts, as shown in Equation (5):(5)$${\mathrm{lnE}}_{\mathrm{it}}{\;=\;a}_{\mathrm{it}}{\;+\;\mathrm{blnU}}_{\mathrm{it}}{\;+\;\mathrm{clnGDP}}_{\mathrm{it}}{\;+\;\mathrm{dlnEI}}_{\mathrm{it}}{\;+\;\mathrm{flnTS}}_{\mathrm{it}}{\;+\;e}_{\mathrm{it}}$$
where *i* represents the province and *t* represents the year (*i* = 1, 2, …, 30; *t* = 1996, 1997, …, 2005 or 2006, 2007, …, 2015). The other variables are the same as Equation (2).

It is necessary to select a suitable model based on the characteristics of the panel data. The specific method is as follows. First, the Hausman test is used to determine whether a random-effects model should be used. If the corresponding p-value is far less than the significance level, the fixed-effects model (Equation (6)) or mixed-effects model (Equation (7)) is suitable. Otherwise, a random-effects model (Equation (8)) should be used. Second, the likelihood ratio test is used to determine whether the mixed-effects model or the fixed-effects model should be selected. If the *p*-value of the F-test statistic is less than the significance level, a fixed-effects model should be used. Otherwise, a mixed-effects model should be used:(6)$${\mathrm{lnE}}_{\mathrm{it}}{\;=\;a\;+\;\mathrm{blnU}}_{\mathrm{it}}{\;+\;\mathrm{clnGDP}}_{\mathrm{it}}{\;+\;\mathrm{dlnEI}}_{\mathrm{it}}{\;+\;\mathrm{flnTS}}_{\mathrm{it}}{\;+\;e}_{\mathrm{it}}$$
(7)$${\mathrm{lnE}}_{\mathrm{it}}{\;=\;a}_{\mathrm{it}}{\;+\;\mathrm{blnU}}_{\mathrm{it}}{\;+\;\mathrm{clnGDP}}_{\mathrm{it}}{\;+\;\mathrm{dlnEI}}_{\mathrm{it}}{\;+\;\mathrm{flnTS}}_{\mathrm{it}}{\;+\;e}_{\mathrm{it}}$$
(8)$${\mathrm{lnE}}_{\mathrm{it}}{\;=\;a}_{\mathrm{it}}{\;+\;b}_{\mathrm{it}}{\mathrm{lnU}}_{\mathrm{it}}{\;+\;c}_{\mathrm{it}}{\mathrm{lnGDP}}_{\mathrm{it}}{\;+\;d}_{\mathrm{it}}{\mathrm{lnEI}}_{\mathrm{it}}{\;+\;f}_{\mathrm{it}}{\mathrm{lnTS}}_{\mathrm{it}}{\;+\;e}_{\mathrm{it}}$$

### 2.4. Data Source

Due to the lack of data for Tibet, we analyze the data from 30 provinces in China. Considering the different trends of the variables before and after 2006, the data from 2006 to 2015 are analyzed and compared to the analysis of the data from 1996 to 2005 in order to determine the influence change. The data for *E*, *U*, *GDP*, *TO* and *TO_road_* are obtained from the statistical yearbooks of China and all provinces. The *GDP* is converted to the 1996 price level. Transportation, warehousing, and postal services are considered one industrial sector in China and warehousing and postal energy consumption account for a relatively small proportion of the total energy consumption [33]. Therefore, we use this data as transport energy consumption.

The value of *E*, *U*, *GDP*, *EI*, and *TS* are converted in the three regions from 1996 to 2015; thus, the trends of variables are obtained, as shown in Figure 3 and Figure 4. The converted coefficients are 0.00001, 0.5, 0.01, 1, and 1, respectively. The y-axis represents a relative value of each variable.

## 3. Results

As shown in Appendix B, the data are suitable to analyze in the panel data model. The results of the Hausman test and the likelihood ratio test mentioned above are shown in Table 2 and Table 3. The estimation results are shown in Table 4 and Table 5.

As shown in Table 5, the estimation of all variables is significant at the 5% significance level. As a variable reflecting the level of wealth, *GDP* is one of the most important factor for the increased transport energy consumption in the whole country. This result is in accordance with the findings of Zhang et al. [5]. A 1% increase results in a 0.56% increase in energy consumption. *U* is the second most important factor; a 1% decrease results in a 0.40% reduction in energy consumption. The improvement in the *EI* significantly reduces the energy consumption; a 1% decrease results in a 0.27% reduction in energy consumption. The elasticity of *TS* with regard to the transport energy consumption is 0.28. The regression results and the respective values of variables can be analyzed as sample data to obtain the discipline of the impacts.

### 3.1. Impact of GDP

The total *GDP* is always the largest in eastern region while the smallest in western region, as shown in Figure 3 and Figure 4. The impact of *GDP* on transportation energy consumption has been significantly reduced in the central and eastern regions, while the impact in the western region has increased significantly, showing the feature of high elasticity occurred in the central and eastern regions in the first period. As shown in Figure 4, *GDP* in the western region continues to grow while the growth rate is small, which is also similar to the economic development trend in the eastern region between 1996 and 2005. The positive effects are weaker in the more developed regions.

### 3.2. Impact of Urbanization

As shown in Table 4 and Table 5, the elasticity of urbanization rate has increased in all regions. In 2015, the urbanization rate in the eastern region had reached 65%, while the urbanization rate in the central and western regions was about 50%, which only reached the urbanization level of the eastern region in 2005. The elasticity of urbanization had a positive relationship with its value. That is to say, different impacts may result from the different urbanization stages. 

The process of urbanization is accompanied by the process of industrial structure transformation [34]. In 2015, the proportion of tertiary industries in the western region was 43%, while it was 50% in the eastern region. According to Wang et al. [35], China’s urbanization lags behind industrialization. The secondary industry has a significant impact on the total transportation volume, while the tertiary industry has an impact on transportation quality. The tertiary industry had less flexibility in energy consumption in the transportation sector. Therefore, it is necessary to pay attention to industrial restructuring. Accelerating the development of the tertiary industry and promoting high-quality urbanization is effective.

### 3.3. Impact of Energy Intensity

The impact of energy intensity on transportation energy consumption has increased slightly in the western region, while declined significantly in the eastern and central regions, which is in accordance with the change of the energy intensity shown in Figure 4. The elasticity of energy intensity had a positive relationship with the value of energy intensity. Therefore, reducing energy intensity has a significant effect on transportation energy consumption reduction especially in areas with high energy intensity.

The energy intensity mainly depends on the quality of fuel, the degree of fuel combustion, and the proportion of clean energy used, which are directly related to the technology level of energy conservation and emission reduction in transportation. As shown in Figure 4, the energy intensity in the eastern region has dropped significantly where the development of energy-saving and emission-reduction technologies are at a high level due to sufficient research and development (R&D) funding. In 2015, the R&D expenses in the eastern region reached 713 billion yuan, while it was only 101.35 billion yuan in the western region [36]. In addition, energy conservation and emission reduction policies are not in place due to the limitation of the economic development level.

### 3.4. Impact of Transport Structure

In the eastern region where the transport structure is stable and at a low level, the impact of transport structure is also smaller compared to the first period. In the central and western regions, the transport structure was deteriorating and their elasticities also increased. The elasticity of transport structure has a positive relationship with the value of transport structure. Its impact is significant where road transport occupied a large share.

When the industrialization level is low, the impact of the secondary industry on the total transportation volume is very obvious. When the industrialization level reached a certain stage, the effect of industrialization on transportation may not be the change of total transportation volume, but the optimization of the transport structure. In this period, the tertiary industry began to play a more important role [37]. Obviously, the industrialization level is still not sufficient in the central region and western region. With the improvement of the industrialization level, the transportation energy consumption will further increase.

## 4. Discussion

We obtained the impacts of four variables on transport energy consumption. Different from previous studies, we analyzed the results further to obtain the main factor and give relevant policy recommendations in specific region.

In the eastern region, economy growth had the most significant impact on transportation energy consumption, while its elasticity decreased compared to the analyses from1996 to 2005, indicating that the quality of the economy has improved, but is not yet sufficient. Sustainable economic development and appropriate development goals become vital to reduce transport energy consumption. The impact of urbanization has increased significantly. It is necessary to increase the proportion of tertiary industry and improve the level of urbanization.

In the central region, the urbanization rate increased from 40.4% in 2006 to 52.2% in 2015 and its elasticity also increased. Urbanization has become the main contributor. Therefore, the main task is to improve the efficiency of the transport activity in the city and adjust the industrial structure. The transportation intensity remained stable between 2006 and 2015 and its impact on transportation energy consumption was also weakened compared to the first period. The elasticity of the transport structure was obviously increased. Therefore, the primary measure to reduce transportation energy consumption is optimizing transport structure, rather than reducing energy intensity.

In the western region, along with the economic growth, transportation energy consumption will continue to increase. The inflection point of energy consumption has not yet been reached, which is consistent with the findings of Narayan [38]. Therefore, sustainable economic development should be promoted. The energy intensity in the western region has declined, but it is still at a relatively high level. Unlike the central region, reducing energy intensity is still an effective entry point for energy conservation and emission reduction. The use of high-energy transportation vehicles should be reduced. Investment in the R&D of transport energy-saving technologies must increase.

With the development of the economy and the process of urbanization, the main driving force is transformed from *GDP* to urbanization rate. The urbanization process often leads to a surge in private cars. In the process of urbanization, we should plan the development of different transport modes, optimize the transportation structure, and develop low-energy transportation modes.

Although we conducted the analysis in different regions, we ignored the difference between urban and rural areas or between different cities in the specific region. We will further analyze the factors at a lower aggregation level. In addition, the use of energy has significant impacts on the environment. On the one hand, energy consumption may cause a shortage of energy; on the other hand, energy consumption may cause the emissions of greenhouse gas (GHG). In the above, we estimated the factors affecting the volume of energy. In the future, we will use panel data to further investigate the factors influencing GHG emissions. 

## 5. Conclusions

In this study, the factors affecting transport energy consumption in China’s three regions were determined using the STIRPAT model. The OLS regression analysis for the period 1996–2005 was compared to the analysis of period 2006–2015 based on the panel data of 30 provinces. Regression analysis were also conducted for the eastern, central, and western regions. 

The results show that the impacts of *GDP*, *U*, *EI*, and *TS* on transport energy consumption are all positive. From the perspective of a specific variable, the elasticity of *GDP* has declined in eastern and central regions where the economy level is higher. It has increased in the western region where the economy level is lower. As the level of urbanization increases, its impact on transport energy consumption will further increase. Reducing energy intensity has a significant effect on reducing transportation energy consumption in areas with high energy intensity. The elasticity of *TS* had a positive relationship with the value of transport structure in the central and western regions where road transport occupied a large share as shown in Figure 4.

Finally, we identified the main factor affecting transport energy consumption in all three regions and propose relevant policy recommendations. *GDP* is the main driver of transport energy consumption in the eastern region, while urbanization is the main driver in the other two regions. Along with the economic growth, transportation energy consumption will continue to increase. Sustainable economic development should be promoted in all regions. Moreover, it is necessary to increase the proportion of tertiary industry and improve the level of urbanization. In the central region, the primary measure to reduce energy consumption should shift from reducing transport intensity to adjusting transport structure. In the western region, reducing energy intensity is still an effective entry point. Reducing the use of high-energy transportation vehicles and increasing investment in the R&D of transport energy-saving technologies are also effective.

## Figures and Tables

**Figure 1 ijerph-16-00555-f001:**
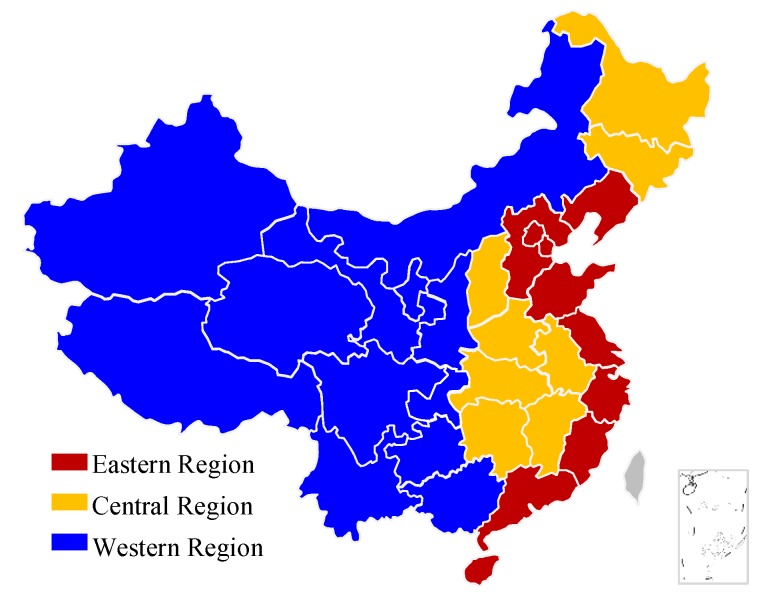
Three regions in China defined by the National Bureau of Statistics.

**Figure 2 ijerph-16-00555-f002:**
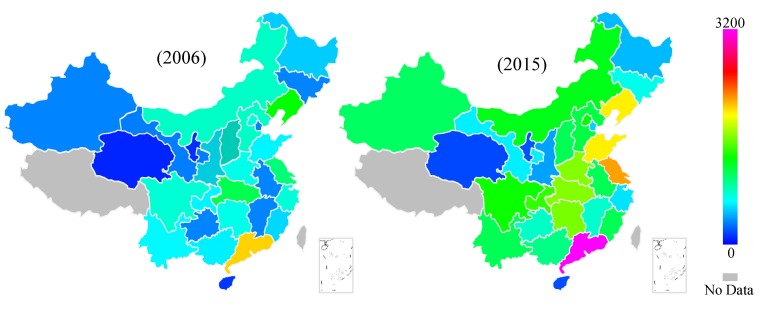
Energy consumption of the inter-provincial transportation sector in 2006 and 2015 (Unit: 10,000 tons of standard coal).

**Figure 3 ijerph-16-00555-f003:**
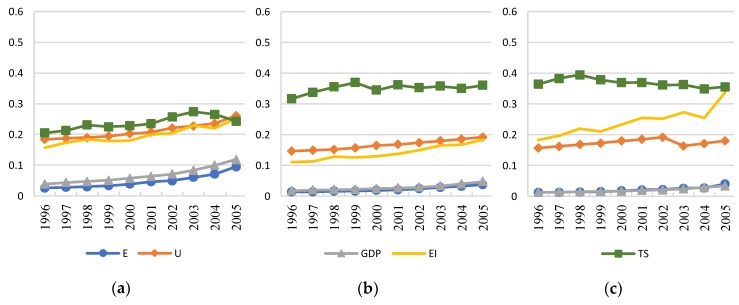
Trends of the variables in three regions (1996–2005). (**a**) Eastern Region; (**b**) Central Region; (**c**) Western Region.

**Figure 4 ijerph-16-00555-f004:**
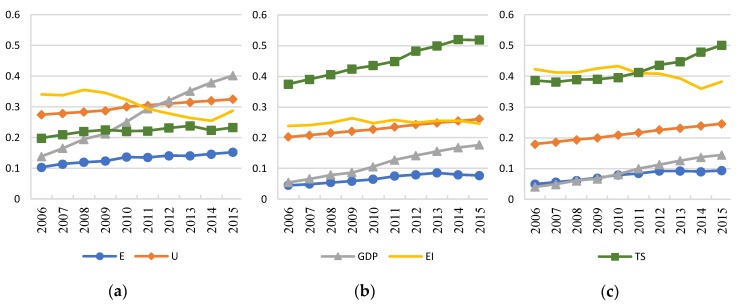
Trends of the variables in three regions (2006–2015). (**a**) Eastern Region; (**b**) Central Region; (**c**) Western Region.

**Table 1 ijerph-16-00555-t001:** Declaration of variables.

Variable Name	Definition	Unit	Literature Support
Energy consumption (*E*)	Total transport energy consumption	10,000 tons of standard coal	Achour et al. [13]; Zhang et al. [14]
Urbanization rate (*U*)	Proportion of urban population to the total population	%	Yang et al. [30]; Lin et al. [17]
Regional gross domestic product (*GDP*)	Regional gross domestic product	Billion Yuan	Liu et al. [15]; Lin et al. [17]; Wu et al. [22]
Energy intensity (*EI*)	Energy consumption per converted ton-km	Kgce/10^4^ ton-km	Lin et al. [17]; Xu et al. [6]; Zhang et al. [14]; Wang et al. [27]
Transport structure (*TS*)	Proportion of road transport ton-km to the total transport ton-km	%	Achour et al. [13]; Zhang et al. [14]; Yang et al. [30]
Transport ton-km (*TO*)	Transport activity which is measured by ton-km	Ton-km	Achour et al. [13]; Zhang et al. [14]; Yang et al. [30]

**Table 2 ijerph-16-00555-t002:** Selection results of panel data model (1996–2005).

Test Type	All Provinces	Eastern Region	Central Region	Western Region
Hausman test	23.04 ***	7.89 *	6.46 *	5.57 *
Likelihood ratio	377.62 ***	-	-	-
Model type	F ^1^	R ^2^	R	R

^1^ indicates a fixed-effects model. ^2^ indicates a random-effects model. * indicates a 10% significance level and *** indicates a 1% significance level.

**Table 3 ijerph-16-00555-t003:** Selection results of panel data model (2006–2015).

Test Type	All Provinces	Eastern Region	Central Region	Western Region
Hausman test	9.11 *	7.99 *	53.57 ***	18.77 ***
Likelihood ratio	26.14 ***	34.53 ***	21.23 ***	79.43 ***
Model type	F ^1^	F	F	F

^1^ indicates a fixed-effects model. * indicates a 10% significance level and *** indicates a 1% significance level.

**Table 4 ijerph-16-00555-t004:** Panel estimation results (1996–2005).

Region	All Provinces	Eastern Region	Central Region	Western Region
Variable	ln*E*	ln*E*	ln*E*	ln*E*
ln*GDP*	0.67 ***	0.77 ***	0.70 ***	0.65 ***
ln*U*	0.18 **	0.16 ***	0.15 ***	0.23 ***
ln*EI*	0.37 ***	0.49 ***	0.72 ***	0.57 ***
ln*TS*	0.19 ***	0.32 ***	0.23 ***	0.15 ***
Constant	0.72 ***	0.34 ***	1.94 ***	0.27 ***
Adjusted R-squared	0.92	0.94	0.97	0.84
F-statistic	866.74	402.62	680.93	148.94
Standard error	27.41	5.17	1.17	11.72

** indicates a 5% significance level. *** indicates a 1% significance level.

**Table 5 ijerph-16-00555-t005:** Panel estimation results (2006–2015).

Region	All Provinces	Eastern Region	Central Region	Western Region
Variable	ln*E*	ln*E*	ln*E*	ln*E*
ln*GDP*	0.56 ***	0.52 ***	0.38 ***	0.93 ***
ln*U*	0.40 **	0.36 ***	0.41 ***	0.99 ***
ln*EI*	0.27 ***	0.32 ***	0.16 ***	0.61 ***
ln*TS*	0.28 ***	0.16 ***	0.33 ***	0.23 ***
Constant	1.96 ***	2.40 ***	3.62 ***	−3.41 ***
Adjusted R-squared	0.99	0.99	0.97	0.99
F-statistic	769.55	524.42	231.47	1269.72
Standard error	4.65	1.61	0.73	0.41

** indicates a 5% significance level. *** indicates a 1% significance level.

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
