# Peer review of "Using Panel Data to Evaluate the Factors Affecting Transport Energy Consumption in China’s Three Regions"

_ijerph, 2019, doi:10.3390/ijerph16040555_

Round 1
Reviewer 1 Report
This paper investigates the factors affecting transport energy consumption in China's three regions using inter-provincial panel data during the period 2006-2015 and 1996-2005. OLS regressions provide elasticity estimates of transport energy consumption with respect to specific variables. For example, the authors find that GDP is the main driver of transport energy consumption in eastern region while urbanization in the other two regions.
Basically, the topic of the research is interesting. However, numerous studies exist in the literature dealing with the relationship between GDP, urbanization etc. and energy consumption (or emissions or any other related variables). The present research should be justified due to its focus on i) China's three regions and ii) transport energy consumption. My impression is that while the paper could generally be reconsidered for publication, it should be improved along the lines pointed out below in order to make it publishable.
The abstract is not appealing! You write "Through the stochastic impacts by regression on population, affluence, and technology (STIRPAT) model, the OLS regression was conducted to investigate the factors affecting transport energy consumption in China's three regions" You should simply write: "... OLS regressions were conducted to investigate the impact of GDP, urbanizatzion, energy intensity and transport structure on transport energy consumption...." Main findings are poorly described in the abstract. Revision is needed. You write "The elasticity and the value of transport structure, energy intensity and urbanization have a positive curve relationship separately." Do you mean something like "The elasticity of transport energy consumption with respect to transport structure, energy intensity and urbanization have a positive and significant relationship.". Do not use the term "curve"!
I do not see Section 2 worth publishing in its present form. Do not review other approaches. You rely on OLS regressions which is standard. Section 2 could be labeled "Methodology and Data."
First, briefly describe your OLS approach.
Second, describe the variables on the left-hand side and the right-hand side of your regression equation LHS: transport energy consumption; RHS: GDP, urbanization, energy intensity and transport structure.
Third, how do you measure these variables (Table 1 and its description must be improved, e.g. what do you mean with turnover???).
Fourth, what tells us the literature as to the impact of GDP, urbanization, energy intensity and transport structure on transport energy consumption, respectively on energy consumption in general or on emissions. As regards this there are numerous studies which are being almost entirely neglected by the authors. For example, the impact of GDP on energy consumption is to some extent related to research on the so called "environmental kuznets curve". The authors neglect this completely.
One study is e.g. Glaeser, E.L., Kahn, M.E., 2010. The greenness of cities: Carbon dioxide emissions and urban development. Journal of Urban Economics 67, 404-418. As summary statistic for all variables should be added.
3. Description of the results is not appealing (see my comment on the Abstract). I suppose Figures 3 and 4 are not based on the regressions. Therefore, I recommend integration into Section 2 . But what is the dimension of the y-axis???
On line 262 you write "An inverted U-shaped curve relationship exists between transport energy consumption and GDP in central region and eastern region." I cannot see this in the data. Please explain!!!
4. Conclusions: Sentences like "the government should guide the residents to choose green transportation modes, increase the investment in railway construction, and gradually increase the proportion of railway transport" are meaningless wisdoms. They should be dropped.
In the end the paper must clearly answer the following questions (in the Abstract, Conclusions etc.):
Do GDP, urbanization, energy intensity and transport structure have a significant positive or negative impact on transport energy consumption
Are the any differences between the three regions
If there are differences, what can we conclude
Author Response
Dear reviewer:
I am very grateful to your comments for the manuscript. These comments are valuable and useful for the paper. According with your advice, we modified the relevant part in manuscript. The following is a point-to-point response to your comments, and responses are in blue.
------------------------------
Comment 1: The abstract is not appealing! You write "Through the stochastic impacts by regression on population, affluence, and technology (STIRPAT) model, the OLS regression was conducted to investigate the factors affecting transport energy consumption in China's three regions" You should simply write: "... OLS regressions were conducted to investigate the impact of GDP, urbanizatzion, energy intensity and transport structure on transport energy consumption...." Main findings are poorly described in the abstract. Revision is needed. You write "The elasticity and the value of transport structure, energy intensity and urbanization have a positive curve relationship separately." Do you mean something like "The elasticity of transport energy consumption with respect to transport structure, energy intensity and urbanization have a positive and significant relationship.". Do not use the term "curve"!
Response:
Thank you for pointing this out. We have rewritten this part based on your suggestion. We have revised “(a) The influencing degree of each variable varies from region to region.” to “First, the changes of the influencing degree in three regions are considered.”. We have revised “Through the stochastic impacts by regression on population, affluence, and technology (STIRPAT) model, the OLS regression was conducted to investigate the factors affecting transport energy consumption in China's three regions.” to “Through the stochastic impacts by regression on population, affluence, and technology (STIRPAT) model, the OLS regressions were conducted to investigate the impacts of gross domestic product (GDP), urbanization, energy intensity and transport structure on transport energy consumption in China's three regions.”. We have revised “The elasticity and the value of transport structure, energy intensity and urbanization have a positive curve relationship separately.” to “The elasticity of transport energy consumption with respect to transport structure, energy intensity and urbanization have a positive and significant relationship.”. Other statements have also been modified.
Comment 2: I do not see Section 2 worth publishing in its present form. Do not review other approaches. You rely on OLS regressions which is standard. Section 2 could be labeled "Methodology and Data." First, briefly describe your OLS approach.
Response:
Thanks for your suggestion. We made corrections based on your comments. Section 2 has been labeled "Methodology and Data.". OLS approach are described briefly at first in the lines 71-75. We have added that “The ordinary least squares(OLS) method finds the best function match of the data by minimizing the sum of squared errors. The least squares estimate is the best linear unbiased estimate of the model parameters.”.
Comment 3: Second, describe the variables on the left-hand side and the right-hand side of your regression equation LHS: transport energy consumption; RHS: GDP, urbanization, energy intensity and transport structure.
Third, how do you measure these variables (Table 1 and its description must be improved, e.g. what do you mean with turnover???).
Response:
Table 1 which contains the description of the variables has been improved. For a simple and clear expression, we use transport ton-km to replace transport turnover. Explanation of transport ton-km has been given, that is transport activity measured by ton-km.
Comment 4: Fourth, what tells us the literature as to the impact of GDP, urbanization, energy intensity and transport structure on transport energy consumption, respectively on energy consumption in general or on emissions. As regards this there are numerous studies which are being almost entirely neglected by the authors. For example, the impact of GDP on energy consumption is to some extent related to research on the so called "environmental kuznets curve". The authors neglect this completely.
Response:
This recommendations are valuable. The impacts of GDP, urbanization, energy intensity and transport structure on transport energy consumption concluded from previous literatures have been discussed. According to your suggestion, we also refer to two papers on the traffic Kuznets curve in the lines 110-113. That is “Xu et al. found that the nonlinear impact of urbanization exhibits an inverted “U-shaped” pattern in China's transport sector. The nonlinear effect of economic growth on CO2 emissions is consistent with the Environmental Kuznets Curve (EKC) hypothesis, whereas inconsistent findings have also been detected.”
Comment 5: One study is e.g. Glaeser, E.L., Kahn, M.E., 2010. The greenness of cities: Carbon dioxide emissions and urban development. Journal of Urban Economics 67, 404-418. As summary statistic for all variables should be added.
Response:
The form to summary statistic for all variables is intuitive. However, the sample data for one variable is up to 600 units, which is hard to be shown in the chart. We provides the data in three regions shown in Figure 3 and Figure 4. If necessary, we can offer these data statistics as appendix.
Comment 6: Description of the results is not appealing (see my comment on the Abstract). I suppose Figures 3 and 4 are not based on the regressions. Therefore, I recommend integration into Section 2 . But what is the dimension of the y-axis???
Response:
Thanks for your suggestions. We have modified the results. We have integrated Figures 3 and 4 into section 2. And we have added the explanation of the y-axis in the line 206-207. “The value of E, U, GDP, EI, and TS are converted in the three regions from 1996 to 2015; thus, the trends of variables are obtained, as shown in figure 3 and figure 4. The converted coefficients are 0.00001,0.5,0.01,1,1 respectively. The y-axis represents a relative value of each variable.”
Comment 7: On line 262 you write "An inverted U-shaped curve relationship exists between transport energy consumption and GDP in central region and eastern region." I cannot see this in the data. Please explain!!!
On line 262 you write "An inverted U-shaped curve relationship exists between transport energy consumption and GDP in central region and eastern region." I cannot see this in the data. Please explain!!!
Response:
These statements in paper are not accurate. We have revised the statements to “The elasticity of GDP has declined in eastern and central regions where the economy level is higher. It has increased in the western region where the economy level is lower.”.
Comment 8: In the end the paper must clearly answer the following questions (in the Abstract, Conclusions etc.):
Do GDP, urbanization, energy intensity and transport structure have a significant positive or negative impact on transport energy consumption
Are the any differences between the three regions
If there are differences, what can we conclude
Response:
We have answered these questions. The impacts and differences are shown in Section 3. We have added the conclusion from the differences. That is “With the development of economy and the process of urbanization, the main driving force is transformed from GDP to urbanization rate. The urbanization process often leads to a surge in private cars. In the process of urbanization, we should plan the development of different transport modes, optimize the transportation structure, and develop low-energy transportation modes.” . The conclusions from the differences of urbanization and transport structure are in the lines 302-303, 308-310.
------------------------------
Thank you very much for the comments and suggestions. We are very grateful for your enthusiastic work and hope that the correction will be approved.
With kind regards
Tianxiang Lv
Reviewer 2 Report
In sections 4 and 5 (Discussion and Conclusion), there are 7 sentences with "government should". Do these demands follow from what was discovered in the article? In equation (2), I see coefficients a, b,c,d and f. These become βit in (4)-(6), so apparently the coefficients may depend on time (maybe I am not understanding something). In Tables 4 and 5, I believe I see the discovered coefficients for lnGDP, lnU, lnEI, lnTS in their contribution to lnE. The discovered coefficients are not universal, and apparently depend on time and place. So (2) is not like a law of physics.
It is not surprising that you find positive coefficients. But the conclusions don't follow from these coefficients. I am an atmospheric scientist by profession, though not specializing in statistics. In the articles for my profession, it would be inappropriate to have so many "government should" in an article like this. The discoveries I see in this article (if any) just do not reveal anything new about "government should".
Author Response
Dear reviewer:
I am very grateful to your comments for the manuscript. These comments are valuable and useful for the paper. According with your advice, we modified the relevant part in manuscript. The following is a point-to-point response to your comments, and responses are in blue.
------------------------------
Comment 1: In sections 4 and 5 (Discussion and Conclusion), there are 7 sentences with “government should”. Do these demands follow from what was discovered in the article? In equation (2), I see coefficients a, b,c,d and f. These become βit in (4)-(6), so apparently the coefficients may depend on time (maybe I am not understanding something). In Tables 4 and 5, I believe I see the discovered coefficients for lnGDP, lnU, lnEI, lnTS in their contribution to lnE. The discovered coefficients are not universal, and apparently depend on time and place. So (2) is not like a law of physics.
Response:
Thank you for pointing this out. “government should” has been removed. The variables and coefficients in equation (4)-(6) has been consistent with those in equation (2). The subscript i and t are removed in equation (1)-(2). Considering if the coefficients depend on time and place, there are three panel data regression models. Therefore, equation (2) is suitable.
Comment 2: It is not surprising that you find positive coefficients. But the conclusions don’t follow from these coefficients. I am an atmospheric scientist by profession, though not specializing in statistics. In the articles for my profession, it would be inappropriate to have so many “government should” in an article like this. The discoveries I see in this article (if any) just do not reveal anything new about “government should”.
Response:
The conclusion is obtained through comparing the coefficients and the value of the variable. Your suggestion is very useful for our paper. “government should” has been removed in the revised paper.
------------------------------
Thank you very much for the comments and suggestions. We are very grateful for your enthusiastic work and hope that the correction will be approved.
With kind regards
Tianxiang Lv
Reviewer 3 Report
The paper in the interesting way discusses the problem of modeling of transport energy consumption (on the example of China’s regions). The paper shows recent studies in the field, which were published in the last years. The presented text meets the formal criteria of scientific studies: the empirical analysis is based on a theory and the findings are based on the conducted analysis, but in my opinion it needs a little effort before acceptance:
In my opinion every model presented in the paper has to be shown in the same way. The Authors should pay attention to the mode of presentation of the STIRPAT Model, extended STIRPAT Model and panel data models. In the two first they use the symbols: a, b, c, d, e to describe the parameters of the models, while in the last one they use: α, β and ξ. I recommend to verify this style of presentation of the parameters and to choose one of them.
I also suggest to use Italic letters for every symbols used in the paper. In my opinion the letters as: I, P and others in the lines: 155-157, 160, 188, 191-194, 243-251 should be presented in this form.
I don’t like the way of presentation of the section 2.2. It starts from the presentation one of the factors of the extended STRIPAT Model (U) but in my opinion it isn’t clear for what this symbol is presented. The same note is related to the explanation of the symbol EI, which is presented for the first time in line 174 but it is described in the line 192. In my opinion the description of this factor should appears earlier (see: description of factor TS in line 175).
I also suggest explaining the differences between the factors: P – in the STRIPAT Model and U – in the extended STIPAT Model. Both of them represent the population, so my question is related with the differences between them.
Please put the attention on the explanation of the symbol: t. In the STRIPAT Model t represents the year but in the panel data model t represents the time dimension. In my opinion it should be presented in the same way and the second one is appropriate. The model can be estimated on the annual data, monthly data and so on, the second form takes into account many different types of data.
After needed revision, paper can be published. Reviewer wishes not to review the revised article.
Author Response
Dear reviewer:
I am very grateful to your comments for the manuscript. These comments are valuable and useful for the paper. According with your advice, we modified the relevant part in manuscript. The following is a point-to-point response to your comments, and responses are in blue.
------------------------------
Comment 1: In my opinion every model presented in the paper has to be shown in the same way. The Authors should pay attention to the mode of presentation of the STIRPAT Model, extended STIRPAT Model and panel data models. In the two first they use the symbols: a, b, c, d, e to describe the parameters of the models, while in the last one they use: α, β and ξ. I recommend to verify this style of presentation of the parameters and to choose one of them.
Response:
Thank you for pointing this out. We have rewritten this part based on your suggestion. The style of presentation of the parameters in section 2.3 has been the same as those in section 2.2.
Comment 2: I also suggest to use Italic letters for every symbols used in the paper. In my opinion the letters as: I, P and others in the lines: 155-157, 160, 188, 191-194, 243-251 should be presented in this form.
Response:
Your opinion is very useful for the paper. We have used Italic letters for every symbols used in the paper.
Comment 3: I don‘t like the way of presentation of the section 2.2. It starts from the presentation one of the factors of the extended STRIPAT Model (U) but in my opinion it isn’t clear for what this symbol is presented. The same note is related to the explanation of the symbol EI, which is presented for the first time in line 174 but it is described in the line 192. In my opinion the description of this factor should appears earlier (see: description of factor TS in line 175).
Response:
The meaning of symbol “U” has been added. “Urbanization in this paper is measured by the proportion of the urban population in total population.” The explanation of the symbol EI et al. has been added in the place occurred first in the lines 103-105.
Comment 4: I also suggest explaining the differences between the factors: P – in the STRIPAT Model and U – in the extended STIPAT Model. Both of them represent the population, so my question is related with the differences between them.
Response:
This proposal is very worthy of revision. We have added the explanation of the differences between the factors P and U in the lines 132-140. “Urbanization in this paper is measured by the proportion of the urban population in total population.” “The population in urban area causes the main part of transport energy consumption. Using total population is not suitable. Therefore, U is used as an influencing factor to reflect the population which influences the environment actually.”
Comment 5: Please put the attention on the explanation of the symbol: t. In the STRIPAT Model t represents the year but in the panel data model t represents the time dimension. In my opinion it should be presented in the same way and the second one is appropriate. The model can be estimated on the annual data, monthly data and so on, the second form takes into account many different types of data.
Response:
This part has been modified. The second one is remained and we have removed the subscript t and i in equation (2) to simplify the discription of STIRPAT model. Because we use annual data in this paper, the subscript t represents the year which replaces the statement “the time dimension”.
------------------------------
Thank you very much for the comments and suggestions. We are very grateful for your enthusiastic work and hope that the correction will be approved.
With kind regards
Tianxiang Lv
Reviewer 4 Report
This paper is highly interesting with regards to the factors that influence energy consumption in China over the period 1996 to 2015 at both an overall level as well as at regional level. My recommendation would to accept the paper subject to minor revision addressing the following points:
- A brief explanation for the choice of the modeling approach (extended STIRPAT)
- Use of the term 'turnover' may be confusing as it may also refer to revenue (when in fact it refers to ton-kms and passenger-km)
- Discussion of the chosen independent variables (especially the measures for energy intensity, EI, and transport structure, TS
- In section 2.4 provide details about the percentage of turnover from warehousing / postal in order to demonstrate that these are not significant.
- Also in section 2.4 provide the source given for turnover conversion coefficients for aviation, rail and road
- Figures 3 and 4 displays the different variables at regional level which is important. It would be relevant also to show the trends for China as a whole.
- It could also be relevant to show these variables in a table to get the exact values?
- A discussion of the obtained elasticity estimates would be relevant (if possible); especially a comparison to elasticity estimates from other countries
- In the conclusion the paper states that road transport has a large market share (it would be useful also to mention whether the share is increasing)
- It could be relevant to outline the perspectives for additional research at a lower aggregation level, to allow comparative analysis between urban vs. rural areas or between different cities.
Author Response
Dear reviewer:
I am very grateful to your comments for the manuscript. These comments are valuable and useful for the paper. According with your advice, we modified the relevant part in manuscript. The following is a point-to-point response to your comments, and responses are in blue.
------------------------------
Comment 1: A brief explanation for the choice of the modeling approach (extended STIRPAT)
Response:
Thank you for pointing this out. We have added the explanation for the choice of STIRPAT model in the lines 86-95. The extended STIRPAT model reflected the advantage of STIRPAT model shown in the lines 90-92, that is “The selection of factors is more comprehensive in the STIRPAT model because the population, affluence, and technology can be decomposed into a number of factors affecting the environment.”.
Comment 2: Use of the term ‘turnover’ may be confusing as it may also refer to revenue (when in fact it refers to ton-kms and passenger-km)
Response:
Your opinion is very useful. For a simple and clear expression, we use transport ton-km to replace transport turnover. Explanation of transport ton-km has been given, that is transport activity measured by ton-km.
Comment 3: Discussion of the chosen independent variables (especially the measures for energy intensity, EI, and transport structure, TS
Response:
This is the part of our paper that is lacking. We have modified this part in the lines 141-159. We have added the measures for energy intensity and transport structure in the lines 170-179.
Comment 4: In section 2.4 provide details about the percentage of turnover from warehousing / postal in order to demonstrate that these are not significant.
Response:
The specific data of turnover from warehousing / postal can’t be obtained caused by the status of China’s statistic. Its small proportion can be demonstrated by many papers which is quoted in the line 204.
Comment 5: Also in section 2.4 provide the source given for turnover conversion coefficients for aviation, rail and road
Response:
The source is presented which is a little far from the occurrence of the coefficients. It is the method proposed by Zhou et al. which is mentioned in the lines 170-171.
Comment 6: Figures 3 and 4 displays the different variables at regional level which is important. It would be relevant also to show the trends for China as a whole.
Response:
Thanks for your suggestion. The trends of the variables combined with the coefficients are considered together to obtain the results. The existing literatures analyzed the impacts from the perspective of the whole country, while the differences of the impacts are not examined between regions. Therefore we showed the trends at regional level.
Comment 7: It could also be relevant to show these variables in a table to get the exact values?
Response:
This makes the paper more intuitive. However, the sample data for one variable is up to 600 units, which is hard to be shown in a table. We provides the data in three regions shown in Figure 3 and Figure 4. If necessary, we can offer these data statistics as appendix.
Comment 8: A discussion of the obtained elasticity estimates would be relevant (if possible); especially a comparison to elasticity estimates from other countries
Response:
This is the part of our paper that is lacking. We have added the impacts obtained from the elasticity of GDP, urbanization, energy intensity and transport structure on transport energy consumption in the lines 96-113, which can be compared with the estimation in this paper.
Comment 9: In the conclusion the paper states that road transport has a large market share (it would be useful also to mention whether the share is increasing)
Response:
Transport structure(TS) is defined as “Proportion of road transport ton-km to the total transport ton-km” . The increasing of the share of road transport can be obtained from the trend of TS in figure 4. We have modified this part. That is “The elasticity of TS had a positive relationship with the value of transport structure in the central and western regions where road transport occupied a large share as shown in Figure 4.”.
Comment 10: It could be relevant to outline the perspectives for additional research at a lower aggregation level, to allow comparative analysis between urban vs. rural areas or between different cities.
Response:
Your suggestion is very useful. We have added this in section 4. That is “Although we conducted the analysis in different regions, we ignored the difference between urban and rural areas or between different cities in the specific region. We will further analyze the factors at a lower aggregation level.”
------------------------------
Thank you very much for the comments and suggestions. We are very grateful for your enthusiastic work and hope that the correction will be approved.
With kind regards
Tianxiang Lv
Round 2
Reviewer 1 Report
---
Reviewer 2 Report
The manuscript has been revised in accord with my simple suggestions.